# A regional scale approach to assess non-residential buildings, transportation and croplands exposure in Central Asia

Chiara Scaini[1], Alberto Tamaro[1], Baurzhan Adilkhan[2], Satbek Sarzhanov[2], Zukhritdin Ergashev[3], Ruslan Umaraliev[4], Mustafo Safarov[5], Vladimir Belikov[6], Japar Karayev[6], Ettore Fagà[7].

[1]National Institute of Oceanography and Applied Geophysics – OGS, Trieste, 34100, Italy
[2]Institute of Seismology, Ministry of Emergency Situations, Almaty, 050060/A15E3F9, Kazakhstan
[3]Tashkent State Transport University, Uzbekistan
[4]Institute of Seismology of Kyrgyz Republic, Bishkek, 720060, Kyrgyzstan
[5]Research Center for Ecology and Environment of Central Asia, Dushanbe, 734063, Tajikistan
[6]Independent consultant, Turkmenistan
[7]RED Risk Engineering Development, Pavia, 27100, Italy

*Correspondence to*: Chiara Scaini (cscaini@ogs.it)

**Abstract**

Critical infrastructure has a paramount role in socio-economic development and its disruption can have dramatic consequences for human communities, including cascading impacts. Assessing critical infrastructure exposure to multiple hazard is therefore of utmost importance for disaster risk reduction purposes. However, past efforts in exposure assessment have predominantly concentrated on residential buildings, often overlooking the unique characteristics of critical infrastructure. Knowing the location, type and characteristics of critical infrastructure is particularly challenging due to the overall scarcity of data and difficulty of interacting with local stakeholders. We propose a method to assess exposure of selected critical infrastructure and demonstrate it for Central Asia, a region prone to multiple hazards (e.g. floods, earthquakes, landslides). We develop the first regionally-consistent exposure database for selected critical infrastructure and asset types (namely, non-residential buildings, transportation and croplands) assembling the available global and regional datasets together with country-based information provided by local authorities and research groups, including reconstruction costs. The method addresses the main known challenges related to exposure assessment of critical infrastructure (i.e. data scarcity, difficulties in interacting with local stakeholders) by collecting national-scale data with the help of local research groups. The analysis also includes country-based reconstruction costs, supporting regional-scale disaster risk reduction strategies that include the financial aspect.

**Short summary**

Central Asia is prone to multiple hazards such as floods, landslides and earthquakes, which can affect a wide range of assets at risk. We develop the first regionally-consistent database of assets at risk for critical infrastructure such as non-residential buildings, transportation and croplands in Central Asia. It combines global and regional data sources and country-based information and supports the development of regional-scale disaster risk reduction strategies for the Central Asia region.

## 1. Introduction

Exposure assessment is the process of collecting information on the type, characteristics and spatial distribution of assets potentially damageable by natural or man-made hazards. Exposure layers are therefore paramount for Disaster Risk Reduction (DRR) as they allow developing strategies to cope with disasters (Nirandjan et al., 2022). Critical infrastructure

plays a paramount role in the risk management cycle, as its failures can exacerbate the impact of disasters (Forzieri et al.,
2018, 2022; Koks, 2022).

Assessing exposure of critical infrastructure is particularly challenging because of their inherent complexity and the difficulty of modeling their mutual interactions (Pant et al., 2018). Many existing global and regional disaster risk models focus on residential buildings or populations, with lesser examples for critical infrastructures, mainly focused on transportation and supply networks (Koks et al., 2019; Argyroudis et al., 2020, Karatzetzou et al., 2022; Mukherjee et al.,
2023). Very few works (e.g. Crowley et al., 2020; Yepes-Estrada et al., 2023) include commercial and industrial buildings, despite their socio-economic relevance for national and global economies and their role in the generation of cascading impacts (e.g. Krausmann and Cruz, 2021). This is partially justified by the incompleteness and inconsistency of existing geospatial information related to critical infrastructure with respect to residential buildings and population data (Batista e Silva et al., 2019). This is one of the reason why critical infrastructure is often modeled through assumptions on
infrastructure density rather than by detailed asset mapping (Koks et al., 2019). Also, once collected, spatial and non-spatial data must be combined to assess exposure of critical infrastructures to single hazards, e.g. for floods (e.g. Fekete et al., 2017, Pant et al., 2018). Such studies often happen at local scale but, in order to be combined into regional and global-scale assessments, there is a strong need for harmonization (Batista e Silva et al., 2019).

The lack of data is not always fulfilled by remote sensing due to the difficulty of identifying some infrastructures (e.g. buried
supply networks), as discussed by Taubenbock and GeiB (2014). To tackle this, it is paramount to access data from national authorities and research institutes who have access to more detailed and reliable information. According to Rathnayaka et al. (2022), establishing a dialogue between stakeholders and the scientific community is a challenge in the development of critical infrastructure exposure databases, and is strongly connected with the difficulty of gathering data, in particular in data-scarce regions. They also highlight the need for establishing a standardized exposure data collection, which is
particularly relevant when assessing exposure to multiple hazards. Multi-hazard exposure taxonomies have been proposed to classify critical infrastructure based on its characteristics (Murnane et al., 2019; Silva et al., 2022) and are particularly relevant in the case of critical infrastructure which is often exposed to multiple hazards that can potentially overlap and interact in space and time (Tilloy et al., 2019). Another limitation of exposure datasets is that they often not include country-based reconstruction costs which are difficult to retrieve in particular for critical infrastructure, limiting the reliability of
financial risk assessment associated to disasters. This is particularly relevant for croplands exposure assessment, whose exposure to floods (Zhang et al., 2023) and drought (Venkatappa et al., 2021) is increasing together with the potential financial losses.

In this study, we present a novel approach to assess exposure of critical infrastructure and support multi-hazard risk assessment. Our method tackles two interrelated challenge identified by the current literature: the difficulty of gathering
country-based data and the lack of dialogue between scientific community and local stakeholders. Exposure data collection was achieved by establishing a dialogue between stakeholders at the regional scale in collaboration with local representatives in the 5 countries of Central Asia, also through dedicated workshops (Peresan et al., 2023). In particular, we gathered country-based reconstruction costs which are commonly difficult to estimate but are paramount to assess financial consequences of disasters and increase financial resilience. The method is demonstrated by assembling the first regionally
consistent exposure database of critical infrastructure for Central Asia based on regional-scale datasets and spatial and non-spatial country-based data for selected critical infrastructure. The exposure dataset is inherently multi-hazards as it includes the characteristics that are deemed relevant for floods, earthquakes and landslides, and potentially useful to assess impact of other phenomena and/or cascading effects. It also includes assets such as commercial and industrial buildings for which no information was available at the time. Data are structured according to the GED4All multi-hazard taxonomy (Silva et al.,
2022), which is here used for the first time in Central Asia to encompass multiple building and infrastructure typologies in a multi-hazard context.

The manuscript describes the study area and all the steps of the exposure assessment methodology including data collection, development of exposure layers and estimation of reconstruction costs of each considered asset type. Finally, we discuss the limitations of the method, its suggested usage and potential improvements,


## 2. Study area

The Central Asia region (Fig. 1) includes 5 countries (Kazakhstan, Kyrgyz Republic, Tajikistan, Turkmenistan, Uzbekistan)
which are diverse in terms of language, currency and socio-economic conditions. Central Asia encompasses a wide variety of climatic areas and geological settings. It is therefore prone to multiple hazards which can affect different parts of the region, including trans-boundary areas (e.g. the Ferghana Valley, where many residential and agricultural activities are located). In particular, floods are increasingly frequent and, in the past, their impacts were often exacerbated by the difficulties related to trans-boundary cooperation, for example in water management (UNECE, 2011; Libert and Trombitcaia, 2015; ). Central Asia is also prone to earthquakes as demonstrated by several regional-scale studies carried out
in the last decades (Ulomov et al., 1999; Bindi et al., 2012; Ullah et al., 2015). Landslides, together with earthquakes and floods, are very frequent in Central Asia and, in the past, were often triggered by natural events such as earthquakes, floods, rainfall and snowmelt (Saponaro et al., 2014; Strom and Abdrakhmatov, 2017). The type and spatial distribution of floods and landslides is also expected to vary due to climate change, which is strongly affecting the region. Another emerging hazard in Central Asia is drought (Zhang et al., 2019) which might affect the region by disrupting productive activities and
exacerbating water management conflicts.

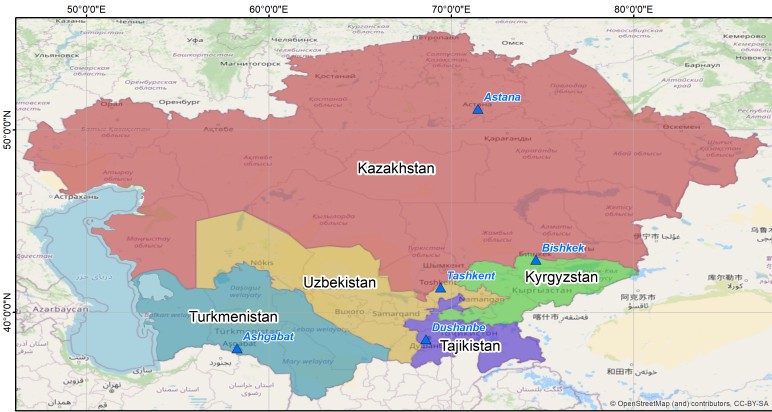

Fig. 1. Map of the 5 Central Asia countries: Kazakhstan, Kyrgyzstan, Tajikitsan, Uzbekistan and Turkmenistan and the corresponding capitals (Astana, Bishkek, Dushanbe, Tashkent and Ashgabat, respectively).


Past exposure assessments in the region were mostly focused on residential buildings (Pittore et al., 2020). However, critical infrastructure is also relevant in the context of Central Asia, and should not be overlooked when performing a comprehensive damage/risk assessment for the region. An effort is therefore required in order to assemble national and regional-scale exposure layers and integrate the available data sources and knowledge, which are currently scattered across
different sources including global databases (e.g. OpenStreetMap) and national-scale aggregated statistics (e.g. national census).

The exposure dataset developed here includes three types of critical infrastructure: non-residential buildings of different types (e.g. commercial, industrial), transportation and croplands. Non-residential buildings comprise workplaces (e.g. industrial sites, commercial buildings), services (e.g. public offices, schools) and other facilities that are extremely relevant

in case of emergencies (e.g. hospitals) and can suffer both physical consequences (e.g. buildings structural damage) and damages, such as the production disruption due to power blackouts and its related financial consequences. The transportation system is  a paramount asset as it enables both the people movement and the transportation of goods across the Central Asia region. Due to its strategic regional and global importance, and has undergone strong changes in the last decades, also in the context of  the Silk Road initiative (Shaikova et al., 2023). Croplands are extremely relevant for the Central Asia economies
as they guarantee both food security and economic development. The primary sector (agriculture, forestry and fishing) accounts for the 26 and 24% of Uzbekistan and Tajikistan GDP, respectively (World Bank, 2020). The share of national GDP in Kyrgyz Republic and Turkmenistan is 14 and 11%, while the lowest value is associated with Kazakhstan (5%). Cotton and cereals (in particular, wheat) are the dominating cropping system in all Central Asia countries (Kienzler et al., 2012). Cotton and wheat, in particular, account for a fraction of cropland area that varies between 30% in Turkmenistan) and
80% in Kyrgyz Republic (FAO, 2019). However, they are threatened by a number of hazardous phenomena, including floods and drought, often exacerbated by climate change and water management issues (Punkari et al., 2014; Li, 2020).
The Central Asia region is therefore characterized by the presence of critical infrastructure exposed to multiple hazards., which can cause multiple impacts yet to be assessed. Despite the relevance of critical infrastructure for the region socio-economic system, and their importance for disaster risk reduction, at the time of the analysis regional-scale exposure datasets
were not available in Central Asia for non-residential buildings, transportation and croplands, and information was scattered across multiple sources. Developing a regional-scale exposure model was therefore required as a first step towards an assessment of potential consequences of floods, earthquakes and landslides that go beyond national boundaries.

### 3. Data collection

The available information on non-residential buildings, transportation system and croplands was collected across the 5 Central Asia countries. The data collection phase was carried out in collaboration with representatives of each country. Data were collected at two different spatial scales, global/regional and national/sub-national, and comprised both official sources and personal communications provided by the manuscript authors and their institutions. Most interaction happened in virtual mode, due to the travel restrictions during the COVID-19 pandemic. The first interactions with participants were carried out
via emails and allowed identifying the type of data that they could provide or gather to develop exposure assessment. After that, dedicated online meetings were periodically organized for each country to discuss specific issues and data requirements, and data were collected through shared folders and tables where each group of partners could contribute. For each country, local partners provided one contact person responsible for data collection, who participated to one initial meeting with all country representatives and one or two country-based, for a total of 7 online meetings held between February and May 2021.
The process was also supported by country-based workshops, whose organization involved the exposure contact person through 2 additional meetings per country, for a total of 10 meetings held between April and December 2021. Exposure workshops provided participants with an overview of the exposure assessment methods to be applied (Peresan et al., 2023) covering all the steps of assembling an exposure development layers for selected study areas using data provided by local partners. This facilitated both data collection and the demonstration of the approaches in a context familiar for participants.
More details are provided by Peresan et al. (2023).

| Category | Type | Global / Regional Data | National Or Sub-National Data | | | | |
| --- | --- | --- | --- | --- | --- | --- | --- |
| | | | Kazakhstan (14 Oblasts) | Kyrgyzstan (7 Oblasts) | Tajikistan (5 Oblasts) | Turkmenistan (5 Oblasts) | Uzbekistan (13 Oblasts) |
| Non residential buildings | Industrial | OpenStreetMap (https://www.openstreetmap.org, 2020) Global mines dataset | Total employed force and percentage employed in industrial sector (https://data.worldbank.org/indicator/SL.TLF.TOTL.IN and https://data.worldbank.org/indicator/SL.IND.EMPL.ZS, from World bank data portal, 2021) | | | | |

| Category | Asset | Global/Regional | Kazakhstan | Kyrgyz Republic | Tajikistan | Turkmenistan | Uzbekistan |
|---|---|---|---|---|---|---|---|
| | | (https://pubs.usgs.gov/of/2010/1255/, Baker et al., 2010) SERA exposure dataset (https://gitlab.seismo.ethz.ch/efehr/esrm20_exposure, Crowley et al., 2020) | | | | | |
| | Commercial | Eurostat employment data (https://ec.europa.eu/eurostat/databrowser/view/LFSQ_EISN2__custom_1304651/default/table?lang=en, last accessed 2022) Eurocommerce employment data (2017) SERA exposure dataset (https://gitlab.seismo.ethz.ch/efehr/esrm20_exposure, Crowley et al., 2020) | Percentage employed force in industrial sector https://data.worldbank.org/indicator/SL.SRV.EMPL.ZS, from World Bank data portal, 2021 | | | | |
| | Education | Schools number and location (https://projectconnect.unicef.org/map/countries, 2020) | Total number of schools in each Oblast (Bureau of National Statistics of the Republic of Kazakhstan, 2018); Schools location shapefile (2018) | School location shapefile; UNICEF school database (2020); Number of schools in each Oblast (2020); School material statistics (World Bank project P149630, Measuring seismic Risk in Kyrgyz Republic', 2015)) | Number of schools in each city; Schools location shapefile (https://geonode.wfp.org, 2018) | Total number of schools in each Oblast (Belikov, V., and Karayev, J., pers. Comm., 2021) | Total number of schools in each Oblast (Ismailov, V., pers. Comm., 2021) |
| | Healthcare | Healthcare facilities database (https://www.healthsites.io/, 2019) | Total number of hospitals in each Oblast (Bureau of National Statistics of the Republic of Kazakhstan, 2018) | Number of hospitals in each city; hospitals Location (http://geonode.mes.kg/, 2020) | Number of hospitals in each city (Institute of water problems, hydropower engineering and ecology, 2020). | Total number of hospitals in each Oblast (Karayev, J., pers. Comm., 2021) | Total number of hospitals in each Oblast (Ismailov, V., pers. Comm., 2021) |
| Agriculture | Crops | Global crop dominance (https://catalog.data.gov/dataset/global-food-security-support-analysis-data-gfsad-crop-dominance-2010-global-1-km-v001, Teluguntla et al., 2015); Global land cover fraction (https://lcviewer.vito.be/download, 2019) | Wheat, cotton and total cereals area, yield production for each Oblast (Bureau of National Statistics of the Republic of Kazakhstan, data for 2020) | Wheat, cotton and total cereals area, yield and production for each Oblast (National Statistical Committee of the Kyrgyz Republic, http://www.stat.kg/, 2020) | Agricultural area for each crop type in each district (Institute of water problems, hydropower engineering and ecology, 2020). | Cotton and total cereals area and production for each Oblast (Belikov, V., and Karayev, J., pers. Comm., 2021) | Wheat, cotton and total cereals area. Yield and production for each Oblast (Ismailov, V., pers. Comm., 2021) |
| Transports | Roads, railways and bridges | OpenStreetMap database (https://www.openstreetmap.org, 02020); Global Roads Inventory Project - GRIP (https://www.globio.info/download-grip-dataset, Meijer et al., 2018) | Description of the transportation network and main highways/railways (Sarzhanov, S., pers. Comm., 2021) | Road maps collected from Caiag geonode (https://geonode.caiag.kg/, 2020); Bridges characteristics (World Bank project P149630, Measuring seismic Risk in Kyrgyz Republic', 2015) | n.a. | Maps and description of road and railway network (Belikov, V., and Karayev, J., pers. Comm., 2021) | Map of main railroads, total length of railroads per type, railway classified by age of construction (Tashkent State Transport University, 2021) |

**Table 1:** Exposure data collected at regional scale and for each country for the considered exposed assets (non residential buildings, agriculture and transports). Data are collected from global/regional databases, national official sources (e.g. governmental agencies) or

were provided directly by local partners and contributors who collected official sources and conveyed the data together with their personal communications. The year for which data were extracted, or up to which the dataset are updated, is also included in the table.

## 3. Methodology

The general method adopted to assemble regional-scale exposure databases relies on two main procedures:
- Spatial disaggregation. Exposure information is often available in an aggregated form (e.g. total value over a region). In such cases, a common method is spatially to distribute the total value using proxies such as population or land use maps. This operation is called spatial disaggregation and is usually performed using Geographical Information Systems (GIS) or spatial analysis libraries (e.g. Gdal, https://gdal.org/).
- Definition of typologies for exposed assets. Exposure assessment requires the definition of asset typologies based on their characteristics (e.g. buildings are classified by material, age, etc.). However, this information might not be available for some exposed assets. In this case, broad typologies can be defined based on information available for parts of the study area and/or for countries outside the study region with similar characteristics. Typologies were then described using the GED4ALL taxonomy (Silva et al., 2022), specifically developed for multi-hazard and risk
assessment purposes.

Following these two principles, we combined the information collected for each exposed asset type (Table 1) to develop exposure layers for non-residential buildings, transportation and croplands. A strong harmonization effort was performed in order to combine all collected exposure data and support regionally consistent risk assessment activities.

**3.1 Non-residential buildings**

Exposure layers were developed separately for each non-residential asset types considered (schools, healthcare facilities, commercial and industrial) based on the data collected in Table 1. For non-residential buildings, few exposure layers were available and there was scarce information on buildings typologies. The definition of typologies was therefore aimed at identifying the main characteristics of non-residential buildings based on two main assumptions:
- The main building typologies in Central Asia defined in the EMCA project (Wieland et al., 2015; Pittore et al., 2020) are considered valid for non-residential buildings. Note that these typologies were also adopted for the development of the residential exposure layer (Scaini et al., 2023).
- In absence of specific country-based information, we used data sources from post-soviet countries, assumed to be similar in terms of past socio-economic context and technical background with regards to construction methods. In
particular, data from the SERA non-residential buildings' exposure layers (Crowley et al., 2020) for the available post-soviet countries (Estonia, Latvia, Lithuania, Moldova) were used, while for the others (Belarus, Ukraine and Russia) data were not available.

Specific methods adopted for each non-residential asset type are described in the following subsections.

**3.1.1 Schools**

School typologies were extracted from a previous UNICEF project in Kyrgyz Republic collected the main exposure characteristics for 1260 schools constituted by 8380 building units surveyed separately. Statistics were performed on the UNICEF layer assuming that each building block is a separate school sample. According to the dataset, all surveyed schools are constituted by load-bearing masonry or precast concrete (80 and 20%, respectively), and the vast majority is found in
rural areas (88%). This is similar to the overall distribution of residential buildings in Kyrgyz Republic, which, according to Pittore et al. (2020), has more than 90% of load-bearing masonry buildings. We assumed that, in absence of specific data for schools in other countries, all Central Asia schools have the same characteristics surveyed in Kyrgyz Republic. Construction

material was therefore defined as a weighted combination of most common school materials in Kyrgyz Republic. Two school typologies were defined (rural and urban) and associated with the most frequent age, area and occupancy value obtained from the UNICEF database for Kyrgyz Republic:

• Urban schools: material: weighted combination of the most common school typologies in Kyrgyz Republic (59% EMCA1, 10% EMCA3, 31% EMCA4); year of construction: 1960-1990; area: 500-1000 m² (average: 750 m²); occupancy: 300 students; taxonomy: UNK + YBET:1960,1990

• Rural schools: material: weighted sum of the most common school typologies in Kyrgyz Republic (56% EMCA1, 22%
EMCA3 and 22% EMCA4); year of construction: 1960-1990; area: 50-500 m² (average: 275 m²); occupancy: 50-200 students (125); taxonomy: UNK + YBET:1960,1990

School structural costs were provided by local partners in each country. The value of 550 USD/m² was adopted in agreement with most data provided, but high discrepancies were found between the cost in Turkmenistan and Kazakhstan (who
provided the highest values, ranging between 2000 and 4500 USD/m²) and Kyrgyz Republic (the lowest, 470 USD/m²). Digital maps of schools were available for Kyrgyz Republic, Kazakhstan and Tajikistan (Table 1). Each point in the spatial dataset was associated with urban or rural school typologies. Urban schools were identified by intersecting them with the urban polygons available from the GRUMP dataset (CIESIN, 2021), while all other schools were considered rural. The location of schools in Uzbekistan and Turkmenistan was not available, but local partners provided the total number of
schools in each Oblast, which were distributed in the GRUMP urban areas (CIESIN, 2021): rural schools were associated with polygons with an area smaller than 20 km².

### 3.1.2 Healthcare facilities

The location of healthcare facilities by type (clinics, hospitals, polyclinic, dentists, doctors, laboratories and pharmacies), last
updated in 2019, is available from the Healthsites database (Weiss et al., 2020). No information was available on the main characteristics (age, material, floor area) of hospitals in Central Asia. Based on the SERA project (Crowley et al., 2020), which provides non-residential buildings exposure data for European countries, we extracted the characteristics of hospitals in post-soviet countries for which the information is available (Estonia, Latvia, Lithuania and Moldova). The average hospital area is 10,000 m² which was assumed valid for all hospitals of Central Asia. Similarly, for clinics, the average area
from the SERA dataset of post-soviet countries was of 1,000 m². As for the material, we assume that the majority of hospitals are reinforced concrete buildings which correspond to the EMCA2 or EMCA3 typologies of the residential buildings classification introduced by Pittore et al., (2020) and refined by Scaini et al. (2023). Clinics and other healthcare businesses (dentists, doctors, pharmacies) were assumed to have a material similar to the residential buildings in each country. Their typology was defined as the weighted combination of the residential building
typologies in each country, based on their fraction, discarding those whose presence is lower than 5%. Other healthcare facilities (dentists, doctors and pharmacies) were assumed to have the same building typologies and reconstruction costs of retail commercial buildings. Their area was estimated as the weighted sum of the areas of the most common residential building typologies in each country. In particular, the floor area was considered for single-family building typologies, while the dwelling area was used for multi-family building typologies, following the same approach used for medium-to-small
retail buildings. Hospital structural reconstruction costs was estimated based on the country-based costs (USD/m2) provided by local partners in each country: an average value of 1.500 USD/m2 was assumed. The replacement cost of hospitals content is assumed to be 150% of the hospitals structural costs following the approach of HAZUS (FEMA, 2021). The other healthcare facilities

(clinics, dentists, doctors and pharmacies) construction and content costs were assumed to be equal to the construction and content cost of the commercial retail building typologies most common in each country.

### 3.1.3 Commercial buildings

Commercial and services buildings, named here as 'commercial', are broadly distinguished into two categories:

- Wholesale and services. Given the lack of specific data for commercial buildings in Central Asia, we assumed that wholesale and services industrial buildings in Central Asia are similar to the post-soviet ones in European countries, obtained from the SERA database (Crowley et al., 2020). A single wholesale and services building typology was defined as the combination of the most common EMCA typologies in the post-soviet countries (namely, EMCA1, EMCA2 and EMCA5 which represent the 26, 37 and 36% of commercial building stock). The average area and occupancy are calculated as the weighed combination of the area and occupancy of the typologies present in the SERA commercial buildings dataset. The so-defined wholesale and services building typology has an average area of 476 m² and the occupancy is 243 people. This is consistent with existing statistics which estimate that wholesale employees are between 10 and 249 employees, but large wholesale firms can employ up to 700 people (OXIRM, 2014).

- Retail buildings, which are assumed to be distributed along residential areas and to have characteristics similar to residential buildings. A single commercial retail typology was defined, in each country, as the combination of the most common residential building typologies in the national building stock. The most common residential building typologies are EMCA1 (masonry) and EMCA4 (adobe) for Kyrgyz Republic, Tajikistan and Turkmenistan with the additional presence of EMCA5 (wood) and EMCA6 (steel) for Kazakhstan. These typologies are low-to-mid rise and encompass a wide range of construction decades, from the '30s until today. Typologies which account for less than 5% of the residential buildings were discarded. The average retail buildings area was estimated as the weighted combination of storey/dwelling area for each building typology. In particular, the floor area was considered for single-family building typologies, while the dwelling area was used for multi-family building typologies. As for the occupancy, in Europe the large majority of retail businesses are micro-businesses employing fewer than 10 people (but there are large retail companies that employ few thousand people, OXIRM, 2014). In this work, we assumed that retail companies accommodate on average 5 people, and we did not account for large retail companies. Structural cost for retail building typologies was computed as the average of structural costs of each EMCA typology weighted by their relative presence in each country obtained from the residential exposure layer developed in Scaini et al. (2023). The content cost is assumed to be equal to the structural cost following the HAZUS inventory technical manual (2021).

Given that no prior information was available on the number of commercial buildings in Central Asia, their number was estimated based on labor market data based on the following indicators:

- Total number of employees in the commercial sector, derived as a percentage of the total labor force for each country (Table 1).

- Total employees in wholesale and retail sector calculated as a percentage over the total employees in the commercial sector activities. To this purpose, values for Europe were used (Eurostat, last accessed 2022).

- Number of retail employees, calculated as a fraction of the total employees in the Total employees in the wholesale and retail sector. According to Eurocommerce (2017), the fraction of employees in the retail sector in 2015 in Europe was 72%, while in post-soviet countries that belong to the EU union (Estonia, Latvia, Lithuania) was 75% (Eurocommerce, 2017). The remaining fraction is associated with wholesale and services.

- Average occupancy of wholesale and services buildings, obtained from the SERA dataset for post-soviet countries. For retail buildings the occupancy was inferred from the European statistics (Eurocommerce, 2017).

The number of commercial buildings was finally estimated by dividing the total employees in the two categories (services wholesale and retail) by the average occupancy of each category.

Wholesale and services and retail buildings were spatially distributed in urbanized areas extracted from the GRUMP dataset (CIESIN, 2021). In absence of additional information on their spatial distribution, they were disaggregated based on population density, so that a higher fraction of buildings was distributed on highly-populated areas. This approach is similar to the one adopted in the SERA project (Crowley et al., 2020). Commercial areas identified in OSM were also inspected but their coverage was deemed insufficient, so the OSM polygons were not used to locate commercial buildings.

### 3.1.4 Industrial buildings

No prior information was available on the number of industrial buildings in Central Asia (Table 1). The number of industrial buildings was then estimated by dividing the employed force by the average buildings' occupancy. The total employed force and the percentage employed in the industrial sector of each country was obtained from the World Bank data portal (Table 1). In absence of country-based or regional-based information, the average occupancy in industrial buildings was inferred from the SERA non-residential buildings' exposure layers (Crowley et al., 2020) for Post-soviet EU countries.

Industrial buildings can belong to more than one EMCA typology. According to the SERA dataset (Crowley et al., 2020), industrial buildings in post-soviet countries are constituted by 31% of load-bearing masonry (EMCA1), 25% reinforced concrete (EMCA2) and 33% steel (EMCA6). Other typologies are present in lower fraction (less than 10%). In absence of specific information, one broad typology was defined as a combination of the three EMCA typologies. Characteristics such as the average area and the structural cost were computed as the average value of the EMCA typologies weighted by their relative fraction in the building stock. An average area of 2013 m² and an occupancy of 35 was obtained. The structural cost for industrial buildings was computed as the weighted average of the costs retrieved for each considered EMCA typology (see Scaini et al, 2023). As for the content, its value is estimated as 150% of the construction cost, following the HAZUS inventory technical manual (FEMA, 2021).

The location of industrial buildings was associated with industrial areas extracted from the OSM database. Areas devoted to mining and other primary sector activities, available from the global mines dataset (Baker, 2010), were removed from the industrial areas. In order to account for the industrial built-up area only, we assumed that half of the industrial area is accommodating buildings. The estimated number of buildings in each country was distributed on the industrial areas identified by OSM, in a number proportional to the polygons' area. The distribution was made so that there is at least one industrial building for each industrial area.

### 3.2 Transportation assets

For each country, roads and railways were extracted from OSM which was found more reliable for the identification of the primary road network with respect to the GRIP database (Global Roads Inventory Project - GRIP, Meijer et al., 2018). Total length of transportation networks (roads and railway) obtained from OSM was compared with data available at national scale for Uzbekistan and Turkmenistan, showing some discrepancies. However, the spatial location of main transportation lines was also compared with non-digital maps of railway lines provided by local partners (e.g., for Turkmenistan) showing an overall good agreement. Roads and railways were then extracted from OSM and classified based on the GED4ALL taxonomy (Silva et al., 2022) which is in its turn based on the OSM classification. Roads were classified into 4 classes: motorway and trunk, primary, secondary and tertiary. Railways were distinguished between high speed and conventional. Roads classified as 'residential', 'service' and 'unclassified' as well as railways tagged as 'subway', 'tram' and 'unknown' were not included in the analysis.

Bridges were extracted from the OSM layer and additional ones were identified by intersecting the primary road layer with other potential obstacle (rivers, motorways and trunks, primary and secondary roads and railways). The bridge typologies were defined based on the data provided by past projects in the region (e.g., 'Measuring Seismic Risk in Kyrgyz Republic',

implemented by the World Bank) and those provided by one of the Uzbekistan local partners (Tashkent State Transport University), which has a deep expertise in the construction of railways and bridges in the region. Since GED4ALL does not provide a taxonomy for bridges but uses OSM taxonomy for roads, we classified bridges based on a custom taxonomy. Two types of bridges were identified:

• Road bridges: In Uzbekistan, 86% of bridges were constructed between 1960 and 1990. Information on bridge material is not available from local partners, but the project 'Measuring Seismic Risk in Kyrgyz Republic' (World Bank project P149630) identified 1500 bridges in Kyrgyz Republic, most of them made of reinforced concrete and steel. We therefore assume that most road bridges (>80%) are constructed between 1960 and 1990 in reinforced concrete or steel.

• Railway bridges are mostly made of reinforced concrete (95% of the total) and they are multi-span; the average length of
span ranges between 12 and 24 m but most bridges are less than 25m long. We assume that these characteristics are common to all railway bridges in Central Asia.

As for costs, no prior official information on transportation assets' reconstruction costs was available. We defined the costs based on country-based information provided by local partners. Given the variability of costs collected, also due to the different soil and construction conditions, we provide both ranges and average values (Table 3 in the results section).

### 3.3 Croplands

The cotton area and yield in each Central Asia country and each sub-national administrative unit (Oblast) was provided by local partners. Such values were used as a starting point for the definition of the exposure layers. The spatial distribution of different croplands was inferred in two steps:

• First, the areas where cotton and wheat are cultivated were inferred from the global crop dominance dataset (Teluguntla et al,. 2015), available at 1-Km resolution. Cotton is associated with class 3 ("Irrigated Mixed Crops"), together with wheat, rice and orchards. Wheat is also found in other classes (1,2,4,5,7), while class 8 was not considered since the wheat fractions is considered negligible with respect to the other crop dominance classes.

• Second, the land cover cropland fraction (Table 1), which has higher resolution (100m), allows discarding cells
with low fraction of cropland coverage.

Having identified the areas where cotton and wheat crops are present, the country-based information obtained for each country and Oblast was distributed spatially. The total cultivated area of cotton and wheat in each Oblast was disaggregated in each 100-m cells, proportionally to the cropland fraction. The taxonomy for croplands corresponds to the one proposed by GED4ALL taxonomy (Silva et al., 2022). In order to assess the expected exposed value, country-based values of yield and
price were used (Table 4 in the results section). Based on the collected information on production and cost, we calculated the exposed value of cotton and wheat croplands in each 100-m cell and the total values per Oblast and country.

## 4. Results

### 4.1 Non residential buildings

Results of the exposure assessment provide the total number of buildings and exposed value for each country and for the considered non-residential building types (Table 2).

**Table 2.** Number of healthcare (hospitals and clinics), schools, commercial and industrial buildings and their corresponding total
reconstruction cost in each Central Asia country and for the entire region (in million USD).

|  | Non residential building types | | | | Total reconstruction costs (million USD) | | | |
|---|---|---|---|---|---|---|---|---|
| Country | Hospitals and clinics | Schools | Commercial | Industrial | Hospitals and clinics | Schools | Commercial | Industrial |

| | | | | | | | | |
|---|---|---|---|---|---|---|---|---|
| Kazakhstan | 768 | 7462 | 848015 | 65838 | 2045 | 2103 | 137700 | 39760 |
| Kyrgyz Republic | 316 | 1260 | 207866 | 21793 | 823 | 355 | 15000 | 11845 |
| Tajikistan | 180 | 858 | 138868 | 13309 | 503 | 242 | 9400 | 7502 |
| Uzbekistan | 804 | 10287 | 1105651 | 118704 | 2274 | 2900 | 204100 | 64517 |
| Turkmenistan | 176 | 1868 | 139425 | 33727 | 268 | 527 | 8100 | 12220 |
| Central Asia | 2244 | 21735 | 2439825 | 253371 | 5913 | 6127 | 368900 | 135844 |

Higher total non-residential buildings reconstruction costs and are found in Kazakhstan and Uzbekistan. The larger reconstruction costs are associated with commercial buildings, followed by industrial buildings. Both are present in larger number with respect to healthcare and school facilities, but have a lower reconstruction cost per building unit. On average,

non-residential buildings account for the 40% of total buildings reconstruction costs estimated in Central Asia, with larger values (up to 50%) in Turkmenistan and values lower than 30% in Tajikistan.

Non-residential building assets were collected in a geospatial database. Figure 2 shows the distribution of education and healthcare facilities in Central Asia. Similar maps can be produced based on the geospatial database developed for other non-residential building types considered during the project.


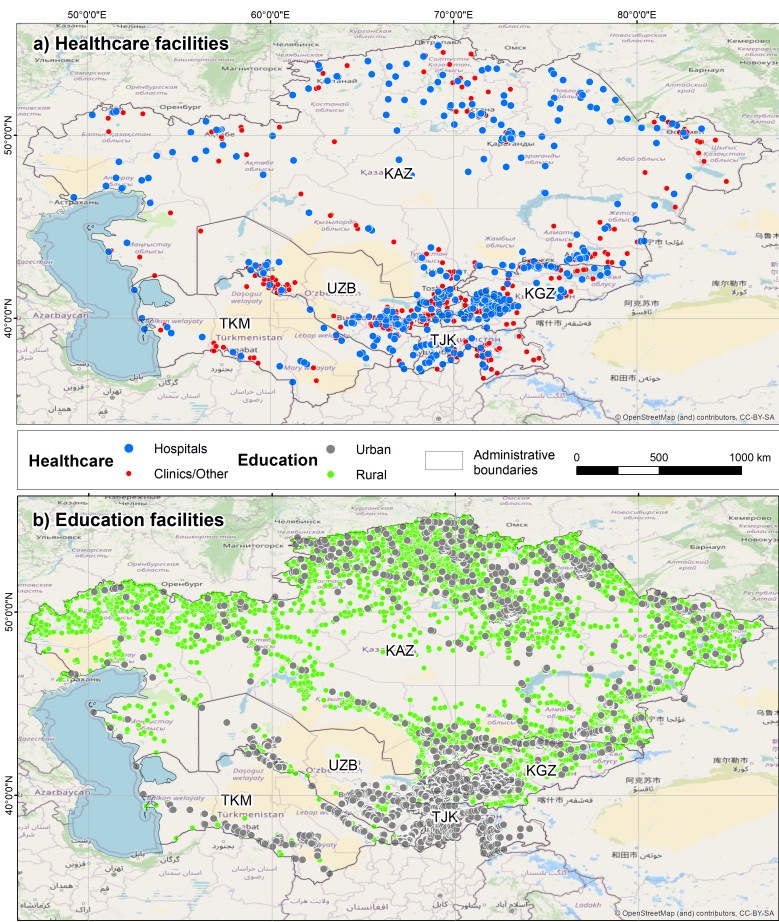

**Figure 2.** Map of healthcare facilities in Central Asia classified into hospitals and clinics and other facilities (top). Map of education facilities classified into rural and urban (bottom). Map data from OpenStreetMap available from https://www.openstreetmap.org (© OpenStreetMap contributors, 2023, distributed under the Open Data Commons Open Database License – OdbL v1.0).


### 4.2 Transportation

Results of the analysis is a geospatial database of the main transportation assets (roads, railways and bridges) in central Asia and the estimation of the associated reconstruction costs. Figure 3 shows the map of transportation assets in Central Asia. Table 3 provides the total length of each type of roads in each country of Central Asia and for the entire region, together with

the total estimated reconstruction costs. Average unit costs for each road type are also provided in the table. The larger reconstruction costs are associated with Kazakhstan, followed by Uzbekistan, and are mostly associated with motorways and highways which have the larger unit cost and a wide coverage in the two aforementioned countries, in particular in Kazakhstan (Fig. 3 and Table 3).

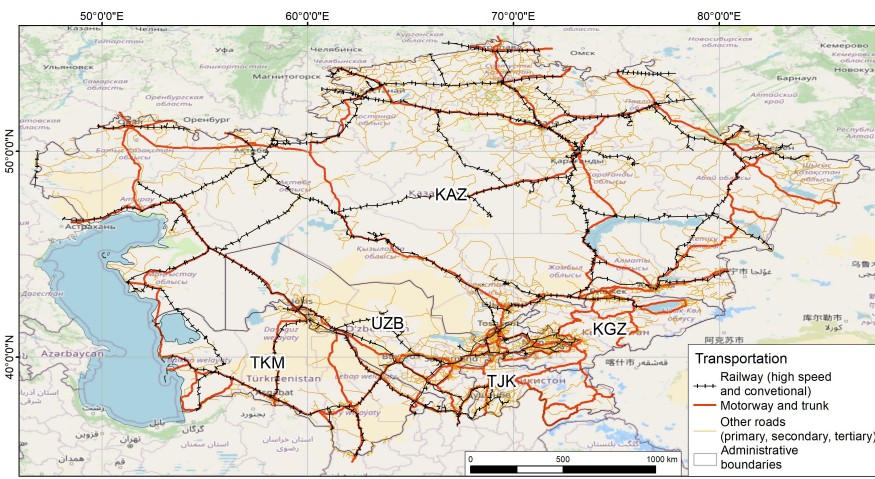

**Figure 3:** Map of the road and railway network in Central Asia included in the exposure database, classified into different types (motorway, trunk, primary, secondary and tertiary). Map data from OpenStreetMap available from https://www.openstreetmap.org (© OpenStreetMap contributors, 2023, distributed under the Open Data Commons Open Database License – OdbL v1.0).

**Table 3**: Total length of road network types and total reconstruction costs estimated for each Central Asia country and for the entire region. Average unit costs for each road type are also provided (third row).

| Country | Road network | | | | Total reconstruction cost (Billion USD) | | |
|---|---|---|---|---|---|---|---|
| | km motorway, highway, trunk | km 1ary | km 2ary | km 3ary | Cost motorway, highway, trunk | Cost 1ary | Cost (all road types) |
| Average unit cost (USD/km) | 2000 | 850 | 500 | 240 | | | |
| Kazakhstan | 17,430 | 8,506 | 19,845 | 46,414 | 34.9 | 7.2 | 63.2 |
| Kyrgyz Republic | 2,787 | 1,996 | 1,878 | 6,578 | 5.6 | 1.7 | 9.8 |
| Tajikistan | 2,645 | 1,014 | 2,856 | 5,539 | 5.3 | 0.9 | 8.9 |
| Uzbekistan | 6,297 | 4,414 | 6,539 | 16,743 | 12.6 | 3.8 | 23.6 |
| Turkmenistan | 6,402 | 1,240 | 1,862 | 7,762 | 12.8 | 1.1 | 16.7 |
| Central Asia | 35,561 | 17,170 | 32,980 | 83,036 | 71.2 | 14.7 | 122.2 |

## 4.3 Croplands

Figure 4 shows the exposure maps produced at regional scale for cotton and wheat croplands at 100-m resolution. Table 4 provides the total wheat and cotton production in each Central Asia Country and Oblast, together with country-based average yield and price. The total exposed value of cotton and wheat croplands for the entire Central Asia region is of approximately 3.000 Million USD. Largest productions of cotton are found in Uzbekistan and Turkmenistan. The greatest production of wheat is found in Kazakhstan, followed by Uzbekistan.

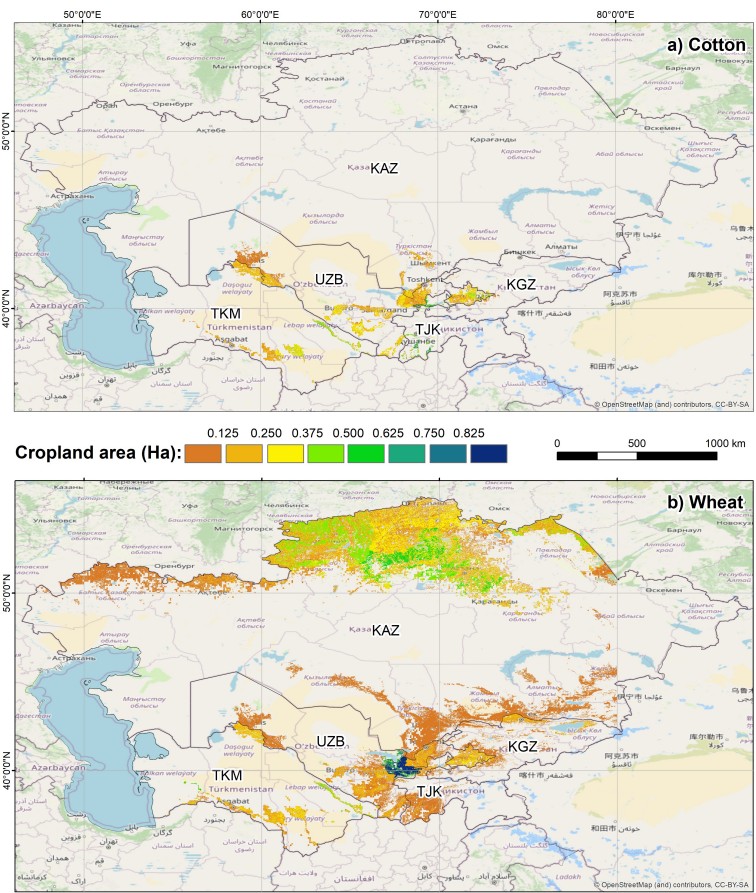

**Figure 4.** Exposure maps produced for cotton (top) and wheat (bottom) croplands at 100m resolution. Map data from OpenStreetMap available from https://www.openstreetmap.org (© Openstreetmap contributors, 2023, distributed under the Open Data Commons Open Database License – OdbL v1.0).

410

**Table 4:** Total area, production and exposed value of cotton and wheat production in each Country and for the entire Central Asia region. Average price and yield are also provided for each country.

| Country | Cotton | | | | | Wheat | | | | |
|---|---|---|---|---|---|---|---|---|---|---|
| | Area (KHa) | Production( Thousand T) | Averag e price (USD/T ) | Average Yield(To ns/Ha) | Total exposed value (Million USD) | Area (KHa) | Productio n(Thousa nd T) | Average price (USD/T) | Average Yield(Ton s/Ha) | Total exposed value (Million USD) |
| Kazakhsta n | 126 | 328 | 304 | 2.6 | 99 | 12142 | 13874 | 91 | 1.4 | 1166 |
| Kyrgyz Republic | 25 | 73 | 600 | 3.3 | 48 | 253 | 629 | 150 | 2.4 | 98 |
| Tajikistan | 146 | 272 | 421 | 0.7 | 43 | 234 | 1416 | 141 | 5.9 | 204 |
| Uzbekista n | 855 | 3094 | 300 | 3 | 757 | 2240 | 7453 | 93 | 6.2 | 1098 |

| | | | | | | | | | |
|---|---|---|---|---|---|---|---|---|---|
| Turkmenis tan | 467 | 1841 | 482 | 2 | 449 | 802 | 1843 | 229 | 2.3 | 422 |
| Central Asia | 1619 | 5608 | 421.4 | 2.32 | 1396 | 15671 | 25215 | 141 | 3.6 | 2988 |

**5. Discussion**

The work presented here develops the first regional-scale exposure layers for selected critical infrastructure assets (namely, non residential building, transportation and croplands). The process of collecting the available information, which is scattered across sources, is particularly challenging for critical infrastructure exposure layers, as also pointed out by Batista e Silva, et al. (2019). Here, we tackled this problem by integrating country-based data into the global and regional datasets used to develop critical infrastructure exposure layers (e.g. OSM, Nirandjan et al., 2022), We collected country-based data for each of the 5 Central Asia countries, thanks to a strong interaction with national research groups and stakeholders (Peresan et al., 2023). The developed approach shows how relevant the contribution of local partners is in developing exposure datasets. It also highlights the importance and difficulties of the data integration process and shows how country-based data can provide an added value to regional-scale exposure datasets of this kind. Country-based data were aggregated and harmonized at the regional scale using an existing multi-hazard taxonomy (Ged4ALL). The method developed here, and exemplified for Central Asia, demonstrates that the Ged4ALL taxonomy is suitable for the development of regional-scale exposure datasets for critical infrastructure. A similar approach can be applied to other regions, but needs to be adapted to the specific conditions (e.g. degree of involvement of national institutions, presence of mobility limitations).

The work is based on several assumptions which are required in order to assemble the first regional-scale layers of their kind. In particular, we assumed that the socio-economic data (e.g. percentage of employees in different sectors) to infer the number of commercial and industrial buildings, as also done by Crowley et al. (2020) for commercial buildings. In our case, due to the absence of specific data on the commercial, industrial and healthcare typologies, we used data from Europe or post-soviet countries assuming that they apply to Central Asia. However, the relative importance of retail and wholesale varies across EU Member States and might vary as well across Central Asia. Hence, further analysis might be required in the future in order to achieve a higher accuracy. Also, we defined broad typologies that comprise multiple building types (e.g. EMCA typologies), as previously done by other authors for buildings (e.g. Wieland et al., 2015 and Pittore et al., 2020 for Central Asia; Calderon et al., 2021 for Central America; Yepes-Estrada et al., 2017 for South America; Yepes-Estrada et al., 2023 at the global scale). These typologies can be associated with multiple vulnerability or fragility curves, combined under general assumptions. For example, retail commercial buildings in Central Asia were assumed to be similar to residential buildings, as also confirmed by local partners during the interaction. Hence, the characteristics of retail buildings were defined based on each country's residential building stock. Different assumptions were performed by Crowley et al., (2020) who developed the first exposure dataset of non-residential buildings for Europe using multiple categories (e.g. classifying commercial buildings into wholesale, retail, offices, hotels and restaurants). The different approaches are mostly due to the larger amount of information available in Europe (e.g. details on building typologies and employment statistics by line of business). Finally, while some non-residential buildings have been mapped by global projects (e.g. schools), information on the spatial distribution of commercial and industrial buildings is usually scarce, as underlined by Batista e Silva et al., 2019 for the European context. Here, they were mapped using a simplified approach based on proxies (e.g. population or land-use), as commonly done in data-scarce regions (De Bono and Mora, 2014; Gomez-Zapata et al., 2023).

Thanks to the high resolution of the population layer adopted in the analysis (Scaini et al., 2023), the exposure dataset for non-residential buildings and croplands was developed on a considerably high resolution (500 and 100m, respectively). This supports the assessment of risk related to floods and potentially landslides, for which a much higher resolution in order to provide reliable results with respect to earthquakes. Nonetheless, regional-scale datasets such as the one presented here can

only support simplified damage/risk assessment that should be calibrated and validated carefully based on past events, when possible, and more specific information on the performance of building typologies considered. This is relevant not only for floods and landslides but also for earthquakes (Wald et al., 2023) to prevent over- or under-estimation of potential risks. For this reason, the suggested usage of the exposure layers provided here (non-residential buildings, transportation and croplands) is limited to the regional or national-scale. However, depending on the type, coverage and quality of data used as input we can associate them with different reliability levels. In particular, the transportation database was developed based on OSM, which is considered a reliable source both in terms of location and classification of roads and railways, and is consistent with the available country-based data. Similarly, the croplands dataset is developed based on recognized products which undergo specific validation processes and national-scale official data (e.g. wheat/cotton production for each oblast). Both datasets are therefore considered reliable for regional-scale damage and risk assessment purposes. Non residential buildings were developed under stronger assumptions and are therefore deemed less reliable. The schools and hospitals layers, despite the availability of location and type for some countries (e.g. from the *healthsites* database, Table 1), rely on scarce information on their characteristics. For this reason, they are considered of medium reliability. Exposure layers for commercial and industrial buildings are developed based on strong assumptions both on the type and distribution of assets, and data integration is required in order to validate the layer. For the time being, their use is suggested as a starting point for further exposure development efforts rather than proper damage/risk assessment. Finally, the dataset of bridges extracted from OSM and identified based on spatial analysis is likely to be incomplete and should not be used to perform a specific risk assessment, but can act as a starting point for the collection of additional information based on complementary surveys and analysis of remote sensing images.

Future work might be required in order to resolve these critical aspects using country-based specific information. In particular, a strong effort should be devoted to validating the dataset based on additional data, which might be available to local public and private stakeholders. This is particularly valid in areas with low data coverage and/or undergoing land use changes. The layers provided here are nonetheless a first step towards Disaster Risk Reduction (DRR) as they provide risk-related information to a broad community of researchers, stakeholders and practitioners, allowing the first-level assessment of expected damages and risks in the region. However, the selection of assets at stake is not limited to the ones considered here, and others might be potentially relevant (e.g. energy production sites and infrastructure). Also, classifications such as GED4ALL (Silva et al., 2022), adopted here, and the one proposed by Murnane et al. (2019) allow for cross-hazard comparisons of risk but do not account for dynamics and feedback loops between the different components of risk (Ward et al., 2022). Future work in this direction might include the estimation of expected risk in the region for one or multiple hazardous phenomena and accounting for potential cascading effects (e.g. flood and drought impacts on croplands and food industry disruption). The time coverage of critical infrastructure exposure data is also dishomogeneous and often incomplete: further efforts should be done in order to update the database in the future, for example using data provided by citizens, not only for buildings (Schorlemmer, et al., 2020; Scaini et al., 2022) but also for other assets such as croplands.

## 6. Conclusions

This work describes an exposure assessment methodology to develop exposure layers for critical infrastructure. This method circumvents the challenges related to the lack of exposure data by collecting country-based information provided by local authorities and research groups, succesfully engaged into a fruitful interaction through meetings and workshops. The method is employed to develop the first high-resolution, regional-scale exposure layers for critical infrastructure in Central Asia for non-residential buildings (healthcare, educational, commercial and industrial), transportation and croplands. The working team collected the characteristics deemed relevant for multiple hazards (earthquakes, floods, landslides) by assembling the available global and regional datasets made available to the scientific community. Reconstruction costs, which are particularly difficult to retrieve, were derived from country-based information for the considered asset types. Results are geospatial layers containing the exposed assets classified using a standardized multi-hazard exposure taxonomy that

supports future multi-hazard and multi-risk assessment. The total exposed value for the different asset types shows that the potential losses associated with non-residential buildings, croplands and transportation are not negligible for financial risk assessment. Exposure database of this kind support further analysis to integrate data from national and sub-national projects
into critical infrastructure datasets and enrich risk-related knowledge towards regional-scale disaster risk reduction strategies.

## Data Availability

The data used to develop the input layer are available at the links provided in Table 1. In particular, the road and railway network was extracted from OpenStreetMap database (https://www.openstreetmap.org) and from the Global Roads
Inventory Project - GRIP (https://www.globio.info/download-grip-dataset) and, for Kyrgyz Republic, from https://geonode.caiag.kg/. The global mines dataset is available at: https://pubs.usgs.gov/of/2010/1255/. Employee statistics were retrieved from the World Bank data portal (https://data.worldbank.org/i) and the Eurostat database (https://ec.europa.eu/eurostat/databrowser) (see Table 1 for details). Healthcare facilities dataset can be downloaded from the Healthsites website (https://www.healthsites.io/), while national data for Kyrgyz Republic can be retrieved at
http://geonode.mes.kg/. The global school dataset was retrieved from the Unicef website (https://projectconnect.unicef.org/map/countries), while national maps are available for Tajikistan ((https://geonode.wfp.org)). Global crop dominance layers can be retrieved at the following link: https://catalog.data.gov/dataset/global-food-security-support-analysis-data-gfsad-crop-dominance-2010-global-1-km-v001, while global land cover fraction was downloaded from https://lcviewer.vito.be/download. National statistics for educational
and healthcare facilities, croplands and transportation were provided by local partners for the purpose of the Strengthening Financial Resilience and Accelerating Risk Reduction in Central Asia (SFRARR) project, but are not publicly available. The spatial layers of exposure for non residential buildings, transportation and croplands developed in this work will be made available at the World Bank data portal (https://datacatalog.worldbank.org/search/dataset/0064117/Central-Asia-Exposure-Data) together with the technical reports developed during the SFRARR project under the Creative Commons Attributions
4.0 license. Data are associated with metadata following the Ged4ALL system (http://riskdatalibrary.org/resources).

## Acknowledgments

The project was developed within the project Strengthening Financial Resilience and Accelerating Risk Reduction in Central Asia (SFRARR), funded by the European Union and implemented by World Bank. We sincerely thank all the project team
members, in particular Dr. Sergey Tyagunov, Dr. Paola Ceresa, Dr. Antonella Peresan, Dr. Gabriele Coccia, Dr. Denis Sandron and Prof. Stefano Parolai and the World Bank specialists, in particular Dr. Stuart Alexander Fraser and Dr. Madina Nizamitdin, for their constructive contribution to the project. We are grateful for the suggestions and feedback from the reviewers, which substantially improved the manuscript. In particular, the input of reviewer 2 played a pivotal role in enhancing the manuscript, ultimately shaping it into its current state.

## Authors contributions

CS, AT, EF developed the exposure assessment methodology, and CS and AT carried out the analyses. All co-authors contributed to the data collection and to the discussion of results. CS prepared the manuscript with contributions from all co-authors.

## Competing interests

The authors declare that they have no conflict of interest.

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
