# Peer review of "A regional scale approach to assess non-residential buildings, transportation and croplands exposure in Central Asia"

_Natural Hazards and Earth System Sciences, 2023_

## Author Comment (AC2)

**ANSWER TO REVIEWER 2:**

*This paper builds a regional exposure database for several types of critical infrastructure in Central Asia, including industrial, commercial buildings, education and healthcare infrastructure, as well as transportation networks (roads and railways) and crops. The dataset is transboundary as it covers the five central Asian countries that were formerly part of the USSR, and is meant to assess damage to several hazard types including flooding, earthquakes, droughts, etc. This database uses a variety of data sources, several of which have to be spatially disaggregated using assumptions that are reasonable and clearly laid out.*

*The writing is clear, the presentation and technical work within are high-quality. Aside from a few minor queries regarding data access and presentation (see below), the major obstacle to publication is a lack of explanation on scientific context and literature aside from the central Asia context. The developed database is multi-hazard, multi-asset, and transboundary: how does that compare with existing databases, e.g., developed for other places? In other words: is the paper just a case-study whose experience is disconnected from that of building multi-layer exposure databases in other regions? Authors should bear in mind the journal's Aims & Scope, which does "not encourage" "localised case studies with no broader implications (in other words, ask yourself, what would someone else in another region learn from the case study that you have done; what is the broader context?)."*

*Thank you for your suggestions. We agree with the reviewer that the manuscript in its current form does not highlight the broader implications of the work and is very focused on the Central Asia context. We re-wrote both introduction and discussion to address this and provide a broader context to the reader.*

*The work presented here is not only a case-study but provides useful insights on how to develop exposure layers at the regional scale based on the combination of both regional-scale data and information at the country level. We focus our analysis on selected critical infrastructures and exposed assets: healthcare facilities, schools, commercial and industrial buildings, transportation network and agricultural system. The database presented here is also intended to be integrated with the residential buildings exposure layer (Scaini et al., this volume). The two papers differ in the method because for critical infrastructures, there were no exposure layers available at the time of the assignment. In fact, before this work, no publicly-available exposure layers of critical infrastructure to multiple hazards existed for Central Asia. Exposure layers for selected assets were developed in some countries (e.g. Kyrgyz Republic) and for selected infrastructure (e.g. transportation) during past projects which are acknowledged in the manuscript. Developing a regional-scale exposure model was nonetheless required as a first step towards an assessment of potential consequences of floods, earthquakes and landslides that go beyond national boundaries. We therefore needed to collect data from different countries and communities and structure them within a regional-scale database, for which we interacted with a wide range of project partners and stakeholders. Gathering data on critical infrastructure is a known challenge, and we include references on how it is usually done, underlining how we interacted with stakeholders, what kind of data was collected and how it was used. The exposure dataset was developed on a considerably high resolution (100m) which supports the assessment of risk related to floods, for which a much higher resolution in order to provide reliable results with respect to earthquakes.*

*This was highlighted in the text, also by broadening the context and the state of the art, in order to clarify the novelty of the work and its validity also for other contexts. The introduction, discussion and conclusions have been rewritten accounting for your suggestions as explained in the following sections.*

*For the paper to fit the journals Aims & Scope, authors need to rethink (and largely rewrite) three sections:*
- *Introduction: authors should review literature on making exposure layers for several types of critical infrastructure: what is considered together and for what reasons? How is their database more comprehensive? Or what obstacles does it overcome that other multi-layer database of critical infrastructure didn't have to deal with? Note this is more than just adding a paragraph to pay lip service to what exists: authors need to review exposure databases for the different layers, the multi-layer efforts, and actively situate this work within this literature, independently from the Central Asian context.*

*The introduction was modified including a literature review on existing methods for the exposure assessment of critical infrastructure as follows. We also point out the challenges associated with developing exposure datasets for critical infrastructures and underline their relevance in the context of Central Asia. We also merged the new part with the old introduction. Part of the new introduction can be found below.*

[revised manuscript text omitted]
. To do that, we use a combination of existing approaches: we merge spatial data with country-based aggregated data to assess exposure of selected critical infrastructures and we subsequently harmonize the dataset at the regional scale. We collect exposure characteristics relevant for multiple hazards, collected by local representatives in the 5 countries of Central Asia establishing a dialogue between stakeholders at the regional scale. The process of data collection and sharing is supported by dedicated workshops (Peresan et al., this volume). Data are then structured according to a recognized taxonomy (GED4ALL, Silva et al., 2022), which is here used for the first time in Central Asia to encompass multiple building and infrastructure typologies in a multi-hazard context. In particular, we included commercial and industrial*

*buildings for which no information was priory available and gathered country-based reconstruction costs to support the assessment of financial consequences of disasters and increase financial resilience.*

*The references contained herein have been also added to the manuscript.*

- *Discussion: it is nice to see authors discuss some of their assumptions there. But these are learning points for other researchers that would want to put together similar databases somewhere else, and for these reasons, the discussion should explain how similar or different the authors' assumptions were from what is done for other exposure databases (and what are reasons that motivated different approaches). In other words: authors need to confront each point they make with the existing literature.*

*The discussion was enhanced by including references to the state of the art and explaining how this work collects existing approaches and/or differs from them. The process of collecting the available information, which is scattered across sources, is particularly challenging for critical infrastructure exposure layers, as also pointed out by Batista e Silva, et al. (2019). Here, we tackled this problem by integrating country-based data into the global and regional datasets used to develop critical infrastructure exposure layers (e.g. OpenStreetMap, Nirandjan et al., 2022), We collected country-based data for each of the 5 Central Asia countries, thanks to a strong interaction with national research groups and stakeholders (Peresan et al., 2023).*

*The work is based on several assumptions which are required in order to assemble the first regional-scale layers of their kind. In particular, we assumed that the socio-economic data (e.g. percentage of employees in different sectors) to infer the number of commercial and industrial buildings, as also done by Crowley et al. (2020) for commercial buildings. In our case, due to the absence of specific data on the commercial, industrial and healthcare typologies, we used data from Europe or post-soviet countries assuming that they apply to Central Asia. However, the relative importance of retail and wholesale varies across EU Member States and might vary as well across Central Asia. Hence, further analysis might be required in the future in order to achieve a higher accuracy. Also, we defined broad typologies that comprise multiple building types (e.g. EMCA typologies), as previously done by other authors for residential buildings (e.g. Wieland et al,. 2015 and Pittore et al., 2020 for Central Asia; Calderon et al., 2021 for Central America; Yepes-Estrada et al., 2017 for South America). These typologies can be associated with multiple vulnerability or fragility curves, combined under general assumptions. For example, retail commercial buildings in Central Asia were assumed to be similar to residential buildings, as also confirmed by local partners during the interaction. Hence, the characteristics of retail buildings were defined based on each country's residential building stock. Different assumptions were performed by Crowley et al., (2020) who developed the first exposure dataset of non-residential buildings for Europe using multiple categories (e.g. wholesale, retail, offices, hotels and restaurants under commercial buildings). The different approaches are mostly due to the larger amount of information available in Europe at national scale. Finally, while some non-residential buildings have been mapped by global projects (e.g. schools), information on the spatial distribution of commercial and industrial buildings is scarce (as also underlined by Batista e Silva et al., 2019 for the European context), and does only support a simplified approach based on proxies (e.g. population or land-use), which is a common approach in data-scarce regions (De Bono and Mora, 2014; Gomez-Zapata et al., 2023).*

*Thanks to the high resolution of the population layer adopted in the analysis (Scaini et al., 2023), the exposure dataset for non-residential buildings and croplands was developed on a considerably high resolution (500 and 100m, respectively). This supports the assessment of risk related to floods and potentially landslides, for which a much higher resolution in order to provide reliable results with respect to earthquakes. Nonetheless, regional-scale datasets such as the one presented here can only support simplified damage/risk assessment that should be calibrated and validated carefully based on past events, when possible, and more specific information on the performance of building typologies considered. This is very relevant, in particular for earthquakes (Wald et al., 2023) to prevent over- or under-estimation of potential risks.*

*All these aspects were integrated into the discussion so that the reader can understand which are the strengths of the method, the assumptions taken, the novel aspects and the limitations to be fulfilled in future work.*

- *Conclusions should summarise in a few sentences what the paper adds to the broader literature.*

*This work contributes to tackling the issues related to exposure assessment of critical infrastructures at the regional scale, while promoting Disaster Risk Reduction in Central Asia by enhancing the availability and sharing of risk-related information. We combine existing global and regional datasets with local-scale data collected thanks to a strong interaction with stakeholders, and include country-based costs that allow for assessing financial risks. We also produce datasets at a high resolution, in particular for crops, which allows to assess risks related not only to earthquakes but also to floods and landslides, for which a higher spatial resolution is required. Conclusions were modified to highlight our contribution and the impact of the work for disaster risk reduction purposes.*

*After that, it could be relevant to spend a bit of time to see whether the new information added to the paper could improve the abstract.*

*The additional information collected during the review and integrated into the manuscript has been included in the abstract.*

*A few queries on data presentation / availability / access:*

*Section 2 text should comment on Table 1 in greater detail. This is true in particular for national and sub-national data. Personal communication sources (institution or public servants) should be mentioned, because local partners must be credited; alternatively, a clear explanation should be provided as to why they cannot be named. The number of oblasts per country should be given to give a better idea of the granularity of the data.*

*Thank you for the comment. The data collection was indeed a pivotal part in the project. Additional challenges were put by the COVID-19 pandemic that negatively conditioned the interactions, with only virtual meetings and no possibility to interact in person. The local research groups, for which the representatives are co-authoring the manuscript, were in charge of gathering reliable information at the country level. They provided it through official documents and/or information from various sources, sometimes collated into personal communications. Dedicated online meetings were periodically organized for each country to discuss specific issues and data requirements, and data were collected through shared folders and tables where each group of partners could contribute. The process was also supported by country-based workshops that provided participants with an overview of the exposure assessment methods to be applied. The process of assembling an exposure development layer was carried out for selected case-study and using data provided from local partners. This facilitated both data collection and the demonstration of the approaches in a context familiar for participants, More details are provided by Peresan et al. (2023). We included more detail on the data provided and the process of data collection. We modified table 1 including the institutions or the persons who provided the information. We also added the number of Oblasts per country to the table.*

*In Table 1, what is missing is a year tag for each data source.*
*We included a year tag to the data sources in Table 1. The year is relative to the last known update of the referenced dataset, as explained in the new table caption.*

*On a related note, it would be good to provide a map of the region including the countries and their names.*
*A map was added showing the Central Asia Region and including each country name.*

*Data availability: is there no way to make the resulting dataset available along with the publication of the paper (rather than to wait for publication by the World Bank)? As things stand, the paper discusses an unpublished database…*
*At the time of the submission, the datasets were in the process of being published. They are now available (since 01/09/2023) under the Creative Commons Attributions 4.0 license at the following link: https://datacatalog.worldbank.org/search/dataset/0064117/Central-Asia-Exposure-Data. The links to the databases and the official project reports were included to the Data availability section.*

---

## Author Response (AR1)

**ANSWER TO EDITOR**

Thank you for the submission of your manuscript "A regional scale approach to assess non-residential buildings, transportation and croplands exposure in Central Asia".

As you know, two reviewers have provided detailed reviews, which you have replied to. Reviewers' recommendations are very different. Referee #1 recommended minor revisions, whereas Referee #2 reported several critical issues in the manuscript and suggested major revisions.

In particular, Referee #2 raised various critical issues about:
• the limited replicability and transfer of the main findings of your study to a broader context, due to the specific features of the investigated case studies;
• the lack of an in-depth review of exposure databases available in literature, independently from the Central Asian context;
• the lack of comparison with similar studies (e.g. different approaches and assumptions adopted for building up exposure databases).

Based on the reviewers' comments and your replies, I would like to invite you to submit an in-depth revision of your manuscript, with substantial improvements mainly in relation to the above-mentioned issues. Please also provide an 'author's reply' to the reviewers and include, in the same author reply document, a track changes document between the old manuscript and the new one.

I look forward to receiving the next version of your manuscript.

*Dear Dr. Bonaccorso, thank you for your comments. We carefully reviewed the manuscript and responded to the issues raised by Reviewer 2, in particular by rewriting the introduction. We included a literature review that goes beyond the study area and highlights the main open challenges in developing exposure datasets of critical infrastructure. We also underline the novel aspects of our methods, compare it with similar studies and discuss its potential for being applied in other study areas. Please find attached my responses to the reviewers (in blue), together with the manuscript and its version in tracking change mode. The line numbers refer to the manuscript with the tracking changes.*

**ANSWER TO REVIEWER 1:**

This reviewer endorses the article for publication with very minor revisions: please add the latitude/longitude grid in the figures, other than the scale bar.
*Thank you for your comment. We've refactored the figures adding the latitude-longitude grid.*

**ANSWER TO REVIEWER 2:**

This paper builds a regional exposure database for several types of critical infrastructure in Central Asia, including industrial, commercial buildings, education and healthcare infrastructure, as well as transportation networks (roads and railways) and crops. The dataset is transboundary as it covers the five central Asian countries that were formerly part of the USSR, and is meant to assess damage to several hazard types including flooding, earthquakes, droughts, etc. This database uses a variety of data sources, several of which have to be spatially disaggregated using assumptions that are reasonable and clearly laid out.

The writing is clear, the presentation and technical work within are high-quality. Aside from a few minor queries regarding data access and presentation (see below), the major obstacle to publication is a lack of explanation on scientific context and literature aside from the central Asia context. The developed database is multi-hazard, multi-asset, and transboundary: how does that compare with existing databases, e.g., developed for other places? In other words: is the paper just a case-study whose experience is disconnected from that of building multi-layer exposure databases in other regions? Authors should bear in mind the journal's Aims & Scope, which does "not encourage" "localised case studies with no broader implications (in other words, ask yourself, what would someone else in another region learn from the case study that you have done; what is the broader context?)."

*Thank you for your suggestions. We agree with the reviewer that the manuscript in its current form does not highlight the broader implications of the work and is very focused on the Central Asia context.*
*The work presented here provides useful insights on how to develop exposure layers at the regional scale based on the combination of both regional-scale data and information at the country level. We focus our analysis on selected critical infrastructures and exposed assets: healthcare facilities, schools, commercial and industrial buildings, transportation network and agricultural system. For critical infrastructures, at the time of the analysis, no publicly-available exposure layers of critical infrastructure to multiple hazards existed for Central Asia. Exposure layers for selected assets were developed in some countries (e.g. Kyrgyz Republic) and for selected infrastructure (e.g. transportation) during past projects which are acknowledged in the manuscript. Developing a regional-scale exposure model for the selected assets was nonetheless required as a first step towards an assessment of potential consequences of floods, earthquakes and landslides that go beyond national boundaries. We collected data from different countries and communities and*

*structure them within a regional-scale database, for which we interacted with a wide range of project partners and stakeholders. The exposure dataset was developed on a considerably high resolution (100m) which supports the assessment of risk related to floods, for which a much higher resolution in order to provide reliable results with respect to earthquakes. Gathering data on critical infrastructure is a known challenge, and we include references on the state of the art, underlining how we interacted with stakeholders, what kind of data was collected and how it was used. We also discussed the caveats of the methods and its potential application to other study areas and/or future work for enhancing it. These aspects were highlighted in the text, also by broadening the context and the state of the art, in order to clarify the novelty of the work and its validity also for other contexts.*

*We re-wrote the entire introduction and substantially modified the discussion to address this and provide a broader context to the reader. The introduction, discussion and conclusions have been rewritten accounting for your suggestions as explained in the following sections. The line numbers refer to the manuscript annotated with tracking changes.*

*For the paper to fit the journals Aims & Scope, authors need to rethink (and largely rewrite) three sections:*
- *Introduction: authors should review literature on making exposure layers for several types of critical infrastructure: what is considered together and for what reasons? How is their database more comprehensive? Or what obstacles does it overcome that other multi-layer database of critical infrastructure didn't have to deal with? Note this is more than just adding a paragraph to pay lip service to what exists: authors need to review exposure databases for the different layers, the multi-layer efforts, and actively situate this work within this literature, independently from the Central Asian context.*

*The introduction was completely rewritten (lines 39-81) including a review of the current methods to develop exposure assessment of critical infrastructure and pointing out the main challenges (lines 44 to 70). We also include a broader justification of why our dataset is addressing these challenges by including spatial and non-spatial data from multiple country-based sources and considering assets exposed to different hazards (lines 71-81).*
*All the information related to the regional context (description of the Central Asia region and hazard and exposure characteristics) was moved to a dedicated section. A figure for the study areas was also added (lines 82-144 and fig. 1).*

- *Discussion: it is nice to see authors discuss some of their assumptions there. But these are learning points for other researchers that would want to put together similar databases somewhere else, and for these reasons, the discussion should explain how similar or different the authors' assumptions were from what is done for other exposure databases (and what are reasons that motivated different approaches). In other words: authors need to confront each point they make with the existing literature.*

*The discussion was enhanced by including references to the state of the art and explaining how this work collects existing approaches and/or differs from them. In particular, we underline why this approach is novel and what is its potential for being adopted in other contexts (lines 442-452). We also compare our assumptions to those of similar works done by other authors (lines 455-473). We also describe the potential of the high-resolution dataset to support flood risk assessment (lines 474-480). We also give emphasis to the use of an existing taxonomy, the GED4ALL, and its advantages and disadvantages for multi- and cross-hazard analyses (lines 446-447 and 509-511). Finally, we point out future work to be done to further enhance the exposure layers also thanks to citizens (lines 517-519). All these aspects are integrated into the discussion so that the reader can understand which are the strengths of the method, the assumptions taken, the novel aspects and the limitations to be fulfilled in future work.*

- *Conclusions should summarise in a few sentences what the paper adds to the broader literature.*

*Conclusions were modified to highlight our contribution and the impact of the work for disaster risk reduction purposes. In particular, we highlighted the fact that we engaged local partners to collect country-based data including reconstruction costs, and that we produce multi-hazard exposure datasets using a standardized taxonomy (lines 524-529).*

*After that, it could be relevant to spend a bit of time to see whether the new information added to the paper could improve the abstract.*
*The additional information collected during the review and integrated into the manuscript has been included in the abstract. In particular, we underline the current gaps and challenges in exposure assessment of critical infrastructure and emphasize our contribution in tackling them ((lines 19-20 and 24-26)*

*A few queries on data presentation / availability / access:*

*Section 2 text should comment on Table 1 in greater detail. This is true in particular for national and sub-national data. Personal communication sources (institution or public servants) should be mentioned, because local partners must be credited; alternatively, a clear explanation should be provided as to why they cannot be named. The number of oblasts per country should be given to give a better idea of the granularity of the data.*
*Thank you for the comment. We enhanced the data collection section by adding information on how data collection was performed (lines 166-174). The data collection was indeed a pivotal part in the project. Additional challenges were put by the COVID-19 pandemic that negatively conditioned the interactions, with only virtual meetings and no possibility*

*to interact in person. The local research groups, for which the representatives are co-authoring the manuscript, were in charge of gathering reliable information at the country level. They provided it through official documents and/or information from various sources, sometimes collated into personal communications. Dedicated online meetings were periodically organized for each country to discuss specific issues and data requirements, and data were collected through shared folders and tables where each group of partners could contribute. The process was also supported by country-based workshops that provided participants with an overview of the exposure assessment methods to be applied. The process of assembling an exposure development layer was carried out for selected case-study and using data provided from local partners. This facilitated both data collection and the demonstration of the approaches in a context familiar for participants, More details are provided by Peresan et al. (2023). We included more detail on the data provided and the process of data collection. We modified table 1 including the institutions or the persons who provided the information. We also added the number of Oblasts per country to the table.*

*In Table 1, what is missing is a year tag for each data source.*
*We included a year tag to the data sources in Table 1. The year is relative to the last known update of the referenced dataset, as explained in the new table caption.*

*On a related note, it would be good to provide a map of the region including the countries and their names.*
*A map was added showing the Central Asia Region and including each country name (Fig. 1).*

*Data availability: is there no way to make the resulting dataset available along with the publication of the paper (rather than to wait for publication by the World Bank)? As things stand, the paper discusses an unpublished database…*
*At the time of the submission, the datasets were in the process of being published. They are now available (since 01/09/2023) under the Creative Commons Attributions 4.0 license at the following link: https://datacatalog.worldbank.org/search/dataset/0064117/Central-Asia-Exposure-Data. The links to the databases and the official project reports were included to the Data availability section.*

---

## Author Response (AR2)

RESPONSE TO EDITOR

Dear Chiara Scaini and co-authors,

your revised paper received an overall positive evaluation by the sole reviewer appointed.

However, he suggested specifying in detail the advancements your work brings about the state of the art of exposure assessment (regardless of place).

Nonetheless, I invite you to further improve the manuscript by addressing this relevant comment.

I'm looking forward to receiving your revised manuscript.

Best regards,

Brunella Bonaccorso

Dear Dr. Bonaccorso,

We revised the manuscript following the reviewer's suggestions and giving more emphasis to the novelty of the approach and how it contributes to addressing the most common challenges in exposure assessment of critical infrastructure.

We thank you and the reviewer for the suggestions that substantially improved the manuscript.

Thank you very much,

Chiara Scaini, on behalf of all co-authors.

RESPONSE TO REVIEWER

This is my second time reviewing this paper. I was reviewer #2 the last time around.

I want to commend authors for a thorough revisions of their paper, that goes most of the way towards addressing my main concerns. The introduction in particular does a good job of explaining the challenges in the existing literature.

A few minor comments aside (see below), my main remark is that no specific conclusions are drawn from the introduction, in terms of the gap that this paper addresses through the great work done by the authors in Central Asia. A couple of sentences stating with precision how the work advances the state of the art of exposure assessment (regardless of place), in the abstract, the introduction and in the conclusions, would help the readers grasp very clearly the innovation the paper brings about (right now they still have to guess or know the state-of-the-art themselves). Formulating the novelty of their work clearly would also help the authors when they build on their work, be it in presenting the work at conferences, in finding new collaborators or in preparing their next grant. Last but not least, this would help the paper to unambiguously meet the journal's aims and scope.

A few minor points below.
Introduction:
Line 56, "region" should be plural.

Line 66: the "aforementioned challenges should be explicitly named, rather than letting the reader guess from the above paragraphs.
It would be best to finish with a few sentences presenting the layout of the rest of the paper.

Section 3, line 131: "online meetings were periodically organised" it would help readers better appreciate the scale of authors' efforts to be a little more precise. For instance, consider stating how many meetings in total / per Central Asian country, and over which time frame (e.g., number of months).

Discussion.
Line 410: "is" should be moved after "the contribution of local partners", not before.

Lines 413-414: "and demonstrate its applicability to this case-study" authors should reformulate what they mean here, probably in a separate sentence.

We thank the reviewer for his comments and suggestions, which improved a lot the manuscript. In particular, his comments were very helpful in identifying the novelty of the work and underlining it through the manuscript. Our main changes are summarized below with line numbers referring to the manuscript with tracking changes.

Following his advice, we included a sentence in the abstract, introduction and conclusions to clearly state the novelty of the work and how it contributes to advances regardless the study area:

- The abstract has been reshaped to focus more on the methodological contributions (in line with the paper) and less on the specific case study (lines 15-25).

- The final part of the introduction states the novelty of the approach emphasizing how it tackles the main challenges due to lack of data and dialogue with local scientific community and stakeholders (lines 72-84). The last sentences describe the outline of the paper to help the reader through it (lines 90-92).

- Conclusions were modified to emphasize the results from the methodological point of view (lines 500-503 and 506-507).

We applied the minor edits required to the text and provided more details on the periodic meetings organized with local partners, which are a total of 17, of which 7 specific for data collection and 10 developed during the organization of country-based exposure workshops (lines 148-156). The process of organizing workshops focused on each country and using local data facilitated the data collection process, as discussed here and in Peresan et al., (2023).

Finally, we clarified the sentence in line 413-414 explaining that the work developed here demonstrates that Ged4ALL can be used to develop regional-scale exposure datasets of critical infrastructures (lines 436-438).

We wish to thank the reviewer for his help in giving a broader perspective to our work and demonstrating its relevance for the academic community.